# How Do We Know Co-Created Solutions Work Effectively within the Real World of People Living with Dementia? Learning Methodological Lessons from a Co-Creation-to-Evaluation Case Study

**DOI:** 10.3390/ijerph192114317

**Published:** 2022-11-02

**Authors:** Grahame Smith, Chloe Dixon, Rafaela Neiva Ganga, Daz Greenop

**Affiliations:** 1Faculty of Health, Liverpool John Moores University, Liverpool L2 2ER, UK; 2Faculty of Business and Law, Liverpool John Moores University, Liverpool L3 5UG, UK

**Keywords:** Living Lab, dementia, methodology, evaluation, reablement, co-creation

## Abstract

Living Labs (LL) are a novel and potentially robust way of addressing real-life health challenges, especially within the dementia field. Generally, LLs focus on co-creating through implementing the quadruple helix partnership as a user-centric approach to co-creating. In the context of this paper, the users were people with dementia and their informal carers. LL are not necessarily environments that evaluate these co-created innovations within the real world. Considering this disconnect between co-creation and real-world evaluation, this paper, as a critical commentary, will reflect on the methodological lessons learnt during the development of an LL model aimed at addressing this discrepancy. The LL at Liverpool John Moores University (LJMU) was commissioned to co-create and then evaluate a new Dementia Reablement Service. The case study findings revealed that the Dementia Reablement Service had a positive impact on the quality of life of people with dementia, suggesting that the service is a catalyst for positive change. In addition, the critical learning from this case study highlights the potential role of LLs in seamlessly co-creating and then evaluating the co-created solution within the real world. A benefit of this way of working is that it provides opportunities for LLs to secure access to traditional research funding.

## 1. Introduction

The Living Lab (LL) at Liverpool John Moores (LJMU) was founded through the work of the Innovate Dementia Project, funded by Interreg Northwest. The aim of this LL was to co-create solutions to address the everyday challenges of people living with dementia. Over the ten years since the LL was founded, the scope of the LL has widened; however this aim is still relevant when working with people with dementia. LL scholarly articles date back over a decade, highlighting the significant opportunities for the use of the LL approach to effectively address a multitude of health challenges [1,2]. Specifically, this approach is viewed as a novel way of addressing real-life health challenges, co-creating sustainable solutions which fit user needs [2,3].

There are many definitions of what an LL is, The European Network of Living Labs defines LLs as “user-centred open innovation ecosystems based on a systematic user co-creation approach, integrating research and innovation processes in real-life communities and setting” [4], whilst others have simply labelled them as virtual or physical spaces in which societal challenges are addressed via collaboration [5].

A distinct feature of LLs is the quadruple helix partnership, which allows for representatives from the public sector, academic institutions, companies, and citizens to be involved [6], ensuring end-users are actively engaged in the open innovation process [7,8]. Actively engaging in this way is popular in health and social care due to the emphasis on user-centric involvement and participation [9,10]. 

As of 2022, there are currently one-hundred-and-forty-seven accredited LLs globally, with most being in Europe and only five in the United Kingdom [4,11], including the LJMU LL. This is a health LL, with a special interest in the dementia field. Tackling dementia is one of the greatest challenges facing the world today. Recently, WHO recognised dementia as a public health priority, with around 55 million individuals globally living with dementia [12]. In the UK, there are currently around 900,000 people living with dementia. It has been predicted that 1 million people will be diagnosed by 2025, and this figure is expected to rise to 1.6 million by 2040 [13]. Informal carers, who are usually unpaid relatives and friends of a person with dementia, meet much of the caring costs themselves; however, there has been a greater focus more recently on improving the care and services for people with dementia living in the UK [13]. 

One such service development is the establishment of dementia reablement services, focussing on maximising the cognitive and social functioning of people with dementia [14,15]. In 2014/2015, a dementia reablement service was co-created and established in a region of the northwest of England [16]. Though called a dementia reablement service, its intention was to complement existing services by providing flexible support to people with dementia, assisting them in living well at home for as long as possible [16]. The service, unlike other reablement services, offers a low-level limited-term service delivered by a dedicated support worker who is not a qualified nurse or occupational therapist [17]. The service is underpinned by a social model of reablement, focusing on enabling people with dementia to retain and, in some cases, regain social skills The overall aim of the service is to co-produce individualised living plans, giving the people with dementia and their informal carers the practical tools (technological and non-technological) to build the confidence to take control of their lives [18]. To engender a real-world fit, the service was co-created through an LL methodology [16]. In addition, the LJMU LL was commissioned to support the co-creation of the service and, at the same time, to evaluate the effectiveness of this service. This presented a novel opportunity to validate the effectiveness of a co-created innovation within the real world. Generally, co-creating and then validating a solution in the real world are methodologically understood as two disconnected processes; however, this was an opportunity to “close the loop” and align these two processes [5]. Using a critical practice commentary approach to reflect on the dementia reablement LL work as a case study, the aim of this paper is to discuss the methodological lessons learnt in the journey towards robustly aligning these two separate processes [19].

### Living Lab Case Study

The outcomes from the LL work on the Innovate Dementia project led to the LJMU LL being commissioned to co-create several dementia-related innovations [20]. Much of this work involved supporting small and medium enterprises to co-create solutions tailored for the health and social care market; most of these innovations were memory enabling technologies aimed at enabling people with dementia to live well [21,22,23]. Subsequently, the LJMU LL was commissioned to incorporate this learning into a new dementia reablement service. The commission involved working in partnership to co-create the new service as an innovation and to co-customise or re-purpose other innovations to be used as part of the service delivery. The commissioners of the service, learning from the exemplar work of the Innovate Dementia project, had already started the co-creation process before the LJMU LL was commissioned. The LJMU LL’s role was to complete this process and ensure the methodology was congruent with an LL approach. In addition, LJMU LL was commissioned to conduct an evaluation of the service, viewed as an innovation in the real world. This commissioning model opened up an opportunity to co-create an innovation and then validate the innovation within the real world. The usual activity within an LL is to co-create and then validate an innovation within a “virtual real-world” environment, effectively making it real-world ready [24]. The challenge was selecting an approach that would capture and validate the service within the real world of people with dementia, including understanding its value and impact [25,26]. 

Unlike a traditional evaluation of a health intervention, the evaluation process was continuous, running in parallel with the co-creation phase and the delivery of the service; monthly check-in meetings with the quadruple helix partnership supported this approach, in addition to engaging with the participants during the data collection phase [26,27]. The co-creation phase signed off the new service, including agreeing on which technological and non-technological innovations would be used to support the reablement of people with dementia [17]. The mediating value for the service and the corresponding innovations package was that any innovation should enable people with dementia and their informal carers to live well, thus improving their quality of life [8,13]. This phase was aligned with a participatory action research process, with people with dementia and their informal carers driving this process through the quadruple helix partnership, akin to a citizen approach [27,28]. Agreeing on the intended value of the service with the quadruple helix partnership was the start of the evaluation process. Once the service was being delivered, any learning from the evaluation was looped back into the quadruple helix partnership, and the service was adapted accordingly [26,27].

Placing such a high value on living well is not surprising, as the majority of people with dementia feel socially isolated, experience loneliness or encounter social exclusion [29]. In part, social exclusion can arise from the public being fearful of engaging with people with dementia [30,31]. The consequences of people with dementia being socially isolated can have a negative impact on their condition, the recent pandemic providing increasing evidence of its effects [32]. Engaging positively with people with dementia within their home environment can not only halt cognitive decline; it can also reduce the effects of social isolation [15,17,33]. To combat social isolation, service interventions included support for accessing social and daytime activities, advice-giving and information, and the use of appropriate assistive technologies [16].

## 2. Methodology of the Co-Created Evaluation Process

The evaluation plan and methods were co-developed through LJMU LL’s interactions with the quadruple helix partnership; people with dementia were supported throughout to be co-equal partners [33]. Baseline and outcome measures were agreed upon, including how the data could be collected as a core part of the service delivery. Building the research capacity within the service was an essential part of co-developing a sustainable evaluation approach. To support capacity building, the LL staff undertook the roles of mentors, meeting regularly with staff through the check-in meetings to encourage them to reflect on the progress of the evaluation, what worked, and what did not [16,34]. To ensure the ethical conduct of the project, university ethics approval was sought and granted [16]. People with dementia were involved throughout as participants and quadruple helix partners and throughout the quadruple helix processes as co-researchers. Consent to participate or not to participate in any part of the project, including the evaluation activities, was respected. 

It is good practice to involve people with dementia in the both the co-creation and evaluation processes, especially in light of dementia reablement being a promotor of social inclusion [15,35]. Admittedly, there are challenges and risks when involving people with dementia in research. These challenges were managed by agreeing on who could participate and who could not. This agreement was signed off on by the university’s ethics committee. People with dementia who were not invited to participate in the evaluation included people with dementia who were deemed too vulnerable, unwell, or as lacking capacity [16,36]. In addition, and as a supportive measure, people with dementia were offered the choice of being interviewed with another person present, either their informal carer or their formal carer (dementia reablement support worker).

To capture the real world of people with dementia, it was decided that the evaluation methodology would use mixed methods [16,37]. The rationale was that a combined empiric and naturalistic approach would capture a more complete data picture [16]. The data collected included:A validated quality of life (QOL) measure, DEMQOL [38], was used pre-, mid-, and post-service implementation.Anonymised narrative data from the support plans—written plans of action were undertaken.Narrative data from open-ended interviews.

Due to the service providing individualised packages of support, which were only summarised in the support plans, it was decided to provide richer data to utilise an open-ended interview approach, which was facilitated by a researcher [16]. 

## 3. Results

In keeping with a critical commentary narrative, the results section will reflect the holistic data picture [19]. The main reflective cue relates to the following question: what do the data tell us about the impact of the service on the quality of life of people with dementia and their informal carers?

The empiric data indicated that, during the evaluation period of ten months, there were a total of 513 referrals to the service, with 2664 face-to-face contacts with people with dementia and their informal carers [16]. Using DEMQOL measures alone indicated that the QOL of people with dementia stayed the same or declined slightly [16].

The naturalistic data from the support plans highlighted that the priorities of people with dementia and their informal carers were being met. Positive impacts upon QOL included improved social functioning, greater confidence in remembering medication, and an increase in knowing from where and whom to seek help [16]. The following summary of the interview data generally supported these findings.

The new service has been a catalyst for change for some people with dementia. The role of the worker facilitated this change and supported the goals of people with dementia to regain independence and hope. The main challenge for people with dementia was to stay at home for as long as possible and to not feel that a nursing home admission is inevitable. The emphasis of the new service on reablement fits this need for hope and a renewed sense of purpose. Practical help included help with bills and legal processes; it also included being supported in engaging with wider social activities. Goal setting in the real world was important, such as still aiming to go on holiday and travel abroad. This does not mean that the support worker goes on holiday; it means that they provide encouragement while the people with dementia and their informal carer put their holiday plans into action. Having a reablement focus reframes dementia from “being the end of the world” to “there is still opportunity to dream, plan, and achieve”. 

The support worker’s role as a practical “signposting” resource was important, as was their ability to personalise this support within the real world of the people with dementia. For example, assistive technology had a place in the lives of people with dementia; however, it had to fit their real-world needs. The support worker would give the people with dementia and their informal carer the time and space to try out different technologies. This involved testing different assistive technologies within the home setting. This personalised interaction between the support workers and people with dementia and their informal carers ensured that technologies, at a personal level, were only adopted if they had a real-world fit. 

Creating a supportive space for people with dementia and their informal carers was critically important. People with dementia can become frustrated when daily tasks become difficult. This does not mean that they will not continue to strive to perform those tasks; however, feelings of frustration need to be appropriately managed. The additional role of the support worker was to provide this space. People with dementia and their informal carers were given time to talk any frustrations through. It also included supporting the people with dementia in continuing to engage with their social network, with or without their informal carer.

To maintain the voice of the people with dementia involved in the service evaluation, the following narrative insights highlight the impact of the service on their respective QOL. Individuals spoke about their loneliness upon finding out they had dementia: “I was a manager, marketing director of a big hotel, and suddenly I had no life at all…. It was a lonely time, and I was very worried”, but upon joining the service, they described that they “then suddenly [I] had someone to talk to”, highlighting the importance of services like these in allowing people living with dementia to feel part of society again.

Often, people with dementia worry about becoming a burden to their immediate family; however, the individuals from this study discussed how interacting with the Dementia Reablement Service changed their perspective and allowed them to continue living life with hope: “From feeling down in the dumps and worrying about how my wife is going to cope with her mum who lives with us, who has dementia and needs full-time caring, and me diagnosed with vascular dementia. My big worry was ‘how is she going to cope with us?’ Then by sheer luck we met [Dementia Reablement Service Worker], who has turned our lives around. We have now just booked our dream holiday in Japan. I can’t believe how much [Dementia Reablement Service Worker] and the Reablement Team have helped putting us in the right direction and frame of mind”.

Furthermore, another concern centred around having no one available to rely on, especially for those who had no close access to their immediate family: “It was more helpful for me than mum. I was juggling a lot of stuff and just wasn’t coping. I don’t have family. I have no safety net at all. [Turning to mum] It was just you and me, wasn’t it? We’ve got family far away but not anyone who can get involved in things like hospital appointments. I could not have coped without them. End of. Full stop. They recognised straight away that I was spreading myself too thin”. The Dementia Reablement Service enabled people to continue as an independent individual whilst providing the support and information needed to do so.

Similarly, people living with dementia often experience mental health issues: “Literally from the time you first came, we got on well. You helped me so much it was just unbelievable. I felt I had to start something, but I had this black cloud over me. I kept going to the GP. It was just the worst time in my life”; however, the Dementia Reablement service was able to provide the support needed to overcome these obstacles: “You helped me so much”.

## 4. Discussion

The evaluation demonstrated that the co-created service has real-world benefits, and, since the evaluation, the service has been re-commissioned. However, the aim of this paper is to dive deeper and discuss the methodological learning, with an emphasis on critically exploring this learning while demonstrating the lessons learnt. This process is framed by the overall intention of moving nearer to the development of a robust and inclusive approach to co-creating and validating solutions within the real world of people with dementia. The LL approach is a methodological environment which supports the co-creation of innovations that have user-centric value. The challenge is how to evidence this value in the real world, especially where trial methodology is too costly and time-consuming and does not necessarily measure real-world outcomes [26]. The work on the new Dementia Reablement Service was a starting point in addressing these challenges and added value to “closing the loop” and aligning the process of co-creation with the process of real-world validation, leading to the development of a new model of how the LJMU LL works. 

The first phase in this new way of working is to clearly specify the intended value of the innovation during the co-creation process. Within a health context, value could be measured by an improvement in health outcomes and/or quality of life. The intended user-centric value or value proposition of this new service was to support people with dementia in living well. The first lesson was to be more specific about what living well looked like as a shared value. Does a co-created solution improve the social functioning of people with dementia or does it improve the QOL? By being specific, we can then look at how we measure change.

The second phase—again, undertaken during the co-creation process—is to consider how value is measured within the real world of people with dementia and, of course, decide what data are to be collected. Using both empiric and naturalistic methods gave the LL team the methodological latitude needed during the data collection process to fully explore this real world and ascertain the impact of the new service. These data were rich in nature and were both empiric and naturalistic, giving a fuller picture of the impact of the innovation [39]. The importance of taking a more holistic approach to data collection within the dementia field is illustrated by trying to measure QOL. The overall amount of positive change seen in the DEMQOL outcomes appear modest as a trend; these findings are consistent with the literature, where “no change” in the quality-of-life scores or slight improvements are not unusual [40,41]. This trend of modest non-significant improvements in QOL scores is demonstrated in trial-type studies where the intervention group scores are compared to the control group scores [42,43]. The QOL scores within the control group in these types of studies indicate a general pattern of decline in the QOL of people with dementia. On this basis, without the new service, it would be expected that the QOL of people with dementia, who now receive the service, would have declined following a diagnosis of dementia. Without a control group or pre-intervention QOL data, it is not possible to be certain. However, it is probable that the new service helped to arrest decline and perhaps stabilised the QOL of people with dementia receiving the new service [16,40]. 

Co-created innovations within an LL will not always be tested using a study trial approach due to these innovations being introduced quickly into the real world. To measure the impact or to validate the positive impact quickly, there must be the capacity to use all the available evidence [44]. This real-world data will not just be empiric, as human interaction with the innovation also needs to be collected [45]. This data can take the form of narratives or themes generated from narratives. The new service evaluation generated several narrative themes that demonstrated that the service had a positive impact on the QOL of people with dementia [16]. The following is a narrative example of the positive impact [16].

It was not simply emotional support Mr D received, as he described in detail how he was helped with attendance allowance, joining a gym, rehousing, and, most importantly, a system for remembering to take his tablets. As a result of the medication, Mr D’s mood has stabilised, and because of the financial help, he can afford to go to the pub with friends. Even his daughter commented: “You’ve changed so much”, he proudly recalled, adding, “I feel totally capable of looking after myself independently”. Perhaps most profoundly of all, Mr D started to talk about losing his dementia and linked this directly with the intervention of his Dementia Reablement Service support worker.

Using a more holistic approach to data collection ensures that the “declining features” inherent within a diagnosis of dementia are taken fully into account; it also helps in reflecting the participatory nature of the LL approach [28,46]. Which methods are chosen will be dependent on the condition, the innovation, the value proposition, and the real world to be understood. To guide the use of these methodologies, the quadruple helix partnership approach comes into its own, where all relevant parties, including people with dementia, guide the LL team through the exploration of the options available. Combining empiric and naturalistic methodologies generates multiple advantages, with one complementing the other [47]. The case study demonstrates how a lack of real-world sensitivity in measuring QOL in dementia can be counterbalanced by the abundance of rich narrative data [39,48]. However, a lesson learnt from the case study related to the LL team not collecting enough robust empiric data from the period before the introduction of the innovation. Understanding and measuring the real world before and after the introduction of an innovation is vital, especially when looking to apply for traditional research funding.

Methodological sensitivity is not the only consideration when making sense of the real world of people with dementia; ethical sensitivity is equally important. Is the innovation going to improve the lives of people with dementia (good) or will it make things worse (bad)? The new service provided care options around the use of assistive technologies; some of these technologies were co-created by people with dementia, and some were not. The challenge is ensuring that these technologies are fit for purpose and enable people with dementia to live well rather than exacerbating either the symptoms and/or having a negative impact on a person’s QOL [49]. To ensure that a technology is fit for purpose, people with dementia need to be a central part of the co-creation and evaluation of the technology, in a way that is sensitive to their condition and respectful of their needs [50,51]. It must also be acknowledged that people with dementia being involved in an LL project and talking about their condition can be cathartic; it can also support people with dementia in feeling involved and part of society, especially when social isolation occurs [51]. This cathartic process may lead to distress, anger, or sadness; therefore, it is important that the LL team consider different types of contingencies where distress arises [52]. One such contingency is to ensure that formal and informal carers are part of the LL’s quadruple helix partnership and can provide the appropriate support where required.

## 5. Conclusions

Critical learning from the Dementia Reablement Service project has supported the LJMU LL in securing significant funding, both from innovation and traditional research funding. This funding has been used to address the social challenges of people with dementia and other groups of people living with a variety of medical conditions. Innovation funding tends to be focused primarily on co-creating and less on evaluating innovations in the real world. This might relate to co-creation, as an LL process, being generally under-researched and unrecognised as a research methodology in its own right [53]. It might, on this basis, also lack the credibility required for securing traditional research funding. Closing the loop from co-creating to co-evaluating gives the LL process a new dimension and potentially an opportunity to innovate and research at the same time and in a robust manner. However, it is important that, when working with vulnerable groups in the real world, such as people with dementia, they are involved throughout this process in a way that is respectful of the real-world challenges they face on a day-to-day basis. Recently, the LJMU LL has secured traditional research funding, which gives the team the opportunity to further develop its new model of working.

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
