# Peer review of "How Do We Know Co-Created Solutions Work Effectively within the Real World of People Living with Dementia? Learning Methodological Lessons from a Co-Creation-to-Evaluation Case Study"

_ijerph, 2022, doi:10.3390/ijerph192114317_

Round 1

Reviewer 1 Report

In this paper, the authors focused on Living Labs (LL) as a novel and potentially robust way of addressing real-life health challenges, especially within the dementia field. This paper, as a critical commentary, described the methodological lessons learned by the authors, whilst developing a robust validation approach for evaluating co-created solutions within the real-world of dementia. The authors described the LL at Liverpool John Moores University, which routinely offered people with dementia (PWD) access to digital health solutions with a real-world fit.  The authors claimed that this novel Dementia Reablement Service (DRS), positively impacted PWD, suggesting the service is a catalyst for positive change.

The topic is interesting.

My main concern about the paper is the aim and the style of the paper.

Throughout the paper, the authors referred to a technical report published by themselves in 2016: “Greenop, D.T. and Smith, G.M., 2016. Dementia Reablement Service: an evaluation for Cheshire East Council”.

Then, in this paper the authors claimed results which require a thorough knowledge of the published report. For example, in the results, the authors said: “Overall DEMQOL measures indicated the QOL stayed the same or declined slightly [16]”.

In parallel, the authors reported testimonials from participants to highlight the narrative impact of the service. Certainly, these narrative elements are attractive to read, however can we draw conclusions from such particular cases?

For example, in Lines 144-146:  “Literally from the time you first came, we got on well. You helped me so much it was just 144 unbelievable. I felt I had to start something, but I had this black cloud over me. I kept going to the 145 GP. It was just the worst time in my life. You helped me so much”.

The authors must review the articulation between the article and the report and avoid reporting too many narrative elements without subsequent analysis.

My others comments are the following:

Introduction

Line 64-67: The authors said: “The service is underpinned by a social model of reablement rather than a biomedical model of reablement, with the aim to co-produce individualised living plans giving the PWD and their informal carer the practical tools (technological and non-technological) to build the confidence to take control of their lives”.

The authors should explain the term: “a biomedical model of reablement”.

The authors used the term “carer” instead of “caregiver”? Is there a particular reason for that?

What is the definition of a carer and a caregiver, and perhaps the difference between a carer and a caregiver?

Introduction: Living Lab Case Study

Lines 91-92: The authors said: “Unlike a traditional evaluation of a health intervention, the evaluation process was continuous, running in parallel with the co-creation phase and delivery of the service”.

The authors should detail the evaluation process, explaining how this process is continuous: number and frequency of checks, etc.

Methodology of the Co-Created Evaluation Process

Lines 114-115: The authors said: “To support capacity building LL staff undertook the role of mentors, meeting regularly with staff to reflect on the progress of the evaluation, what worked and what did not”.

Same as previous comment, the authors should detail the meetings between the staffs, characterizing the term “regularly”: number and frequency of meetings, etc.

Lines 122-124: The authors said: “Admittedly, there are challenges and risks, these were managed by agreeing with the co-creation group and the ethics committee the exclusion criteria, including not inviting PWD to participate who were deemed vulnerable, unwell or lacking capacity”.

The meaning of the sentence is unclear. Maybe there is a dot missing after the word “committee”.

Lines 125: The authors said: “In addition, PWD were offered the choice of being interviewed with an informal or formal carer”.

The authors should specify what an informal carer is, and what an informal carer is.

Results

Line 135: The authors said: “During the evaluation period there were a total of 513 referrals to the service with  2664 face-to-face contacts with PWD and their informal carers”.

What is the duration of the evaluation period?

Reviewer 2 Report

This is a very interesting paper and has real potential to offer new learning, however, currently it is trying to do too much in a relatively short paper.  The paper's title suggests it is a methodological discussion, presenting learning about how to co-evaluate co-created services but in the paper itself the authors present both methodological discussion alongside the results from the evaluation.  As a result neither aspect of the project is described in sufficient detail to meaningfully contribute to current knowledge.

If the paper's main aim is to present innovation and provide learning in evaluation methodology then significant work needs to be done to develop the description of the methodology and to present a critical discussion of the approach in relation to other evaluation methods.  The methodology described appears to be a fairly straightforward mixed methods approach, utilising three methods of data collection.  It would be helpful to compare and contrast this approach with other mixed-methods approaches to clearly highlight the innovation in this approach.    

The authors list 'anonymised data from the support plans' as one of the data collection methods but no information is provided about what type of data these were, and how they were used in the analysis.

The authors describe the process as 'co-evaluation' but beyond the participants taking part in interviews it is difficult to see, from the information provided, how the evaluation process itself is undertaken in a participatory manner.  Were people with dementia involved in designing the evaluation and in collecting the data?  Or within the data analysis, write up or dissemination?  More clarity is needed about the 'co-evaluation' approach.

In the discussion the authors talk about ethical sensitivity but the remaining part of this paragraph appears to be about ensuring technology is fit for purpose - it would be helpful to draw out in more detail what ethical issues were of importance here and how they were approached within the project.

The paper needs a careful proof read to address missing words and sentences that don't make sense.

The recommendation is for a major revision that would: rebalance the paper and provide more depth and detail on the methodology; more effectively demonstrate the innovation in methods; and fully explain and reflect on the -co-evaluation' approach. 

Round 2

Reviewer 2 Report

I thank the authors for their thoughtful revisions to the paper.  I am happy that all comments and suggestions have been addressed.

One final style point - the use of PWD does not fully recognise the personhood the person with dementia and can contribute to stigma.  However, I would defer to the view of the journal editor on this point.  The use of this acronym may be accepted more widely by the journal. 

Author Response

Reviewer 2:

“One final style point - the use of PWD does not fully recognise the personhood the person with dementia and can contribute to stigma.  However, I would defer to the view of the journal editor on this point.  The use of this acronym may be accepted more widely by the journal”

Acronym has been removed and replaced with “people with dementia”. DRS has also been removed and replaced.